# Discrete-Impulse Energy Supply in Milk and Dairy Product Processing

**DOI:** 10.3390/ma14154181

**Published:** 2021-07-27

**Authors:** Paweł Droździel, Tetiana Vitenko, Viktor Voroshchuk, Sergiy Narizhnyy, Olha Snizhko

**Affiliations:** 1Faculty of Mechanical Engineering, Lublin University of Technology, Nadbystrzycka Str. 36, 20-618 Lublin, Poland; 2Department of Food Technology Equipment, Faculty of Engineering of Machines, Structures and Technologies, Ternopil Ivan Puluj National Technical University, 46001 Ternopil, Ukraine; tetiana.vitenko@gmail.com (T.V.); voroschuk@gmail.com (V.V.); 3Department of Food Technology and Technology Processing of Animal Products, Faculty of Biotechnological, Bila Tserkva National Agrarian University, 09117 Bila Tserkva, Ukraine; sam_nsa@bigmir.net; 4Department of Technologies of Meat, Fish and Marine Products, Faculty of Food Technologies and Quality Management of Products of Agricultural Products, National University of Life and Environmental Sciences of Ukraine, 03041 Kyiv, Ukraine; snezhkoolha@gmail.com

**Keywords:** discrete-impulse energy, hydromechanic, process, dairy products

## Abstract

The efficient use of supplied energy is the basis of the discrete-impulse energy supply (DIES) concept. In order to explore the possibility of using DIES to intensify the hydromechanical processes, the emulsification of milk fat (homogenization of milk, preparation of spreads) and, in particular, the processing of cream cheese masses, were studied. Whole non-homogenized milk, fat emulsions, and cream cheese mass were the object of investigation. To evaluate the efficiency of milk homogenization, the homogenization coefficient change was studied, which was determined by using the centrifugation method, as it is the most affordable and accurate one. To provide the proper dispersion of the milk emulsion, six treatment cycles must be carried out under the developed cavitation mode in a static-type apparatus, here resulting in a light grain-like consistency, and exhibiting the smell of pasteurized milk. The emulsions were evaluated according to the degree of destabilization, resistance and dispersion of the fat phase. On the basis of the obtained data with respect to the regularities of fat dispersion forming in the rotor-type apparatus, the proper parameters required to obtain technologically stable fat emulsion spreads, possessing a dispersion and stability similar to those of plain milk creams, were determined. It was determined that under the DIES, an active dynamic effect on the milk globules takes place. The rheological characteristics of cheese masses were evaluated on the basis of the effective change in viscosity. The effect of the mechanical treatment on the structure of the cheese masses was determined.

## 1. Introduction

The development of energy-saving technologies that meet modern production requirements and provide optimal conditions for the processes is usually based on devising new concepts. The improvement of the most widespread and energy-intensive processes is still an urgent issue for the food industry, in particular the processing of dairy product. Processes such as mixing, homogenization, and emulsification are examples of energy-intensive technologies. As a rule, the intensification of such processes in multicomponent environments takes place due to the supply of external energy, which is introduced into the apparatus by mechanical stirring, the introduction of additional streams of liquid or gas, initiation of vibrations, use of centrifugal forces, acoustic or pulsed influences and powerful electrical discharge. All these methods contribute to the forced relative motion of phases, resulting in the deformation and grinding of the dispersed particles, increasing the period for which the particles stay in the volume of the apparatus and creating a more equal distribution of the dispersed phase in a continuous medium. At the same time, these methods are limited in their effectiveness. Since effective grinding is mainly performed within the energy supply area, the continuous medium and the dispersed inclusions are mixed into a single unit in the volume of the apparatus, and a significant part of the energy is unproductively consumed in order to overcome viscous forces and frictional forces. To use energy effectively, it is necessary to provide a certain level of power for a short period of time (to convert the energy into short powerful pulses). Such energy supplies are referred to in the scientific literature as discrete-impulse energy supplies (DIES) [1,2,3].

The general principles of DIES, its energy and thermodynamic aspects, and the main mechanisms of intensification that are initiated on the basis of this principle have been described previously in detail [2,4,5,6,7]. In the opinion of the authors [8], it is necessary to create a large number of vapor or gas–vapor bubbles in order to implement the conditions for the discrete energy distribution in the working fluid volume of the fluid medium. They can be considered as microtransformers that convert the potential energy accumulated in the system into kinetic energy, which is discretely distributed in space and time. The DIES mechanisms should be used to stimulate hydromechanical processes, or for the purpose of the intensive mixing of multicomponent media, with it being possible to smoothly change the level of intensification of the process within certain limits. In fact, these conditions can be obtained by ultrasonic influence on the liquid [9] or by the creation of special hydrodynamic conditions in the flow [10] in the propellant or rotor apparatus. 

The problem of the intensification of hydromechanical processes in the processing of milk and dairy products is extremely important today. There are different ways of solving this problem. Traditional methods of intensification are based on the concept of local isotropic turbulence, the basis of which is presented in [11] and developed in [12,13]. In accordance with this theory, an important role in stimulating the grinding of hydromechanical particles is played by pulsations of velocity fluctuations in the turbulent flow v’ and p’. The maximum size of the globules formed during grinding in emulsification processes, according to the theory of local isotropic turbulence, is determined from the equation $d_{max}=\frac{\sigma }{\rho_{c}v^{{}^{\prime }2}}$ and is dependent on the coefficient of interphase tension, the values of the turbulent pulsations, and the density of the medium. From an energy point of view, the degree of intensification is dependent on the size of the local power dissipation per fluid mass unit. Therefore, to increase the level of intensification, it is necessary to obtain the maximum values of pulsation velocity. Moreover, the maximum effect occurs in the areas with the highest power dissipation. In order to achieve this, strategies include increasing the speed of the flow in the pipelines, the speed of the mixer rotation, or the roughness of the walls, as well as the use of obstacles, diaphragms, turbulizing gratings, etc. [14,15,16]. It should be noted that, when increasing the amount of energy supplied, the efficiency of its use decreases, as the majority of it is unproductively dissipated through the volume of the continuous phase and to the walls of the apparatus. The authors of [17] explained in detail that it is not appropriate to control the processes of hydromechanical emulsification or homogenization solely on the theory of local isotropic turbulence, since the physical mechanism of the dispersion process is not treated as a direct result of turbulent pulsations. Other researchers [18] have described a mathematical model for the deformation and destruction of globules in liquid and gaseous media based on shear stresses and accelerated flows. 

The principle of shock impact is based on the explosive effect of the cumulative current (the accumulation of energy and its realization in a short period of time in a small space). High values of specific power within the area under treatment are achieved by reducing the spatial and temporal localization of energy, which is possible within a volume of saturated vapor–gas bubbles. In the process of bubble growth or compression in fluid, some non-stationary micro flows, which have a dynamic effect on the dispersed particles, are formed around them. The results presented in [19,20] suggest that the velocity gradient of radial microcurrents around the bubbles is equal to 10^7^ s^−1^, while acceleration exceeds 9.8 × 10^6^ m/s^2^, and the value of the pressure pulse at the boundary with the bubble amounts to 10^9^ Pa, which can cause the destruction of solid dispersions. Among the mechanisms of DIES that we further rely on, we should also mention here cavitation mechanisms which cause the formation of a significant number of vapor–gas bubbles and shear stress effects. These mechanisms can be introduced in a static/dynamic cavitation apparatus, a rotary pulsating apparatus, etc. [21,22,23,24,25,26,27,28,29,30,31,32,33,34,35]. Their use in milk processing technologies is quite promising and important from the point of view of implementing innovative technologies in production.

Let us analyze these mechanisms. It is known that the relative velocity of the dispersion phase is the initial factor causing intensifications in the dispersion medium, and the strength of the hydrodynamic interaction of the particle with it is proportional to the acceleration. Thus, the possibility of crushing the particle is determined by how quickly the velocity of the main phase movement changes. The energy factor of this efficiency mechanism is the velocity of the kinetic energy flow change. Providing such conditions is possible by changing the channel cross-section of a static-type apparatus [36,37,38,39] (Figure 1). The procedure is performed under these conditions, being subjected to the force of the hydrodynamic interaction of the particle with the liquid around each part. The greater the acceleration, the more rapidly the energy transformation will take place, and the larger the proportion of the energy flow that will be spent on the deformation and crushing of parts. In turn, the sharp change in pressure when the liquid passes through the confusion and diffusion areas results in the formation and sufficient growth of vapor bubbles, as well as their successive cavitation buffeting, causing strong dynamic effects.

It has been proved that cavitation can provide the emulsification of two liquids that have not been mixed, on the basis of investigations in early papers by Gazes [40], Chambers [41,42] and Sollner [43]. Investigations [24,44,45] reported that hydrodynamic and acoustic cavitation can be an efficient method for both the inactivation and homogenization of microorganisms in plain milk. Another possibility for affecting the particle dispersion is shear stress, where the velocity gradient is applied in the direction perpendicular to the flow direction. The vectors of the movement velocity of the continuous phase relative to the particle at a point located symmetrically to its center are directed in the opposite directions, and the particle is thus subjected to a tensile force moment, which can cause it to be crushed. This mechanism can be realized through the tangential movement of the liquid in the narrow gap between two coaxial cylinders, the Couette flow [46], under the conditions present in a rotary-pulsating apparatus [32,33,34,35], where the high specific power is distributed uniformly throughout the whole operating volume and is used directly in the discrete zones near each particle. Different authors have paid special attention to the emulsification of different compositions of fat systems [47,48] with different properties [49,50]. At the same time, the stability of the colloid systems, in particular in emulsions with different concentrations and different types of surfactant, is the topic of many investigations [51]. Some authors have contributed to the literature by providing information on the effect of factors such as temperature [52] and pressure [53] in obtaining stable fat dispersions. However, some problems have still not been studied enough in the available scientific and technical literature. This is the state of investigations of the technological operation of the emulsification of milk and oil fats to date, as well as the study of the effect of the above-mentioned factors on its efficiency. Investigations of the system based on the rotor–stator couple for the emulsification of the protein medium in the food industry have been carried out [31]. The distribution of the emulsified products was presented in the paper, as well as an overview of the microstructure of the operating medium. Investigations of emulsification mediums being based on stabilizers such as starch and gum arabic, using the rotor–stator couple, are presented in [32]. The realization of the DIES method, which provides a high density of kinetic energy at discrete points in the volume in order to create the necessary hydrodynamic conditions around each dispersion particle and to provide high-amplitude acceleration values and relative velocity, is similar in these two devices.

We studied the possibility of effectively using equipment operating according to the DIES principle, static cavitation, and rotary-pulsating apparatus in the hydromechanical processes of emulsion and dispersion, with a particular focus on the processing of milk and dairy products. The apparatus design is described in the paper described above [54].

## 2. Materials and Methods

### 2.1. The Object and Methods

Three objects processed in the dairy products industry, for which homogenization and dispersion are mandatory conventional technological operations, were chosen.

These were:-Non-homogenized milk, as the object of hydrodynamic dispersion and homogenization;-Emulsions of milk and oil fats, with a fat phase 3.5–72.5%, as the object of fat emulsification and the investigation of system stability;-Cream cheese mass as the composite protein product originating from cottage cheese, the object of investigation being the emulsification and dispersion efficiency.

The first two objects are fat emulsions with different contents. In terms of its structure, milk is a complicated heterogenous multiphase structure containing a lot of interdependent structural formations. These include the coarse dispersion of milk fat, colloid system of casein particles, lipoprotein particle dispersion, molecular and ionized solutions of the serum proteins, low-molecular nitrogen compounds, milk sugar, salts, etc. [55]. The milk fat in milk consists mostly of separate fat globules independent of each other. Their total amount is 2–6 billion in 1 cm^3^ of milk [56]. It should be noted that in contrast to other food emulsions, milk does not contain oil fats, surface-active substances, or other components that are able to affect the homogenization and dispersion processes. For this reason, in the first stage, it was used as the investigation object to determine the homogenization efficiency under the DIES action [1,2] provided in the static cavitation apparatus.

The second object was the compound food emulsion. In its content (described below), the type of the fat phase was changed, as well as the percentage of surface-active substances. Similar to the first group of investigations, the realization of the DIES mechanism was carried out in a rotor-type apparatus. The emulsification of the milk and oil fats, the regularities of the direct type emulsions, as well as studying their stability and their dependence on the processing parameters were studied.

The third object of investigation was chosen due to its peculiarities. As the cream cheese mass is a pseudo-plastic rheological body, its rheological parameters and structure were investigated in the context of a study quantifying the supplied energy efficiency in the stator–rotor pair.

### 2.2. The Object and Methods for Estimating Homogenized Milk Quality

Whole non-homogenized milk with fat content of 4%, with titrated acidity of 16–20 T, homogeneous consistency, a white color without precipitates and flakes or off-flavor, was used as an object for the research of the cavitation effects on milk fat. To evaluate the efficiency of homogenization, the Gerber method was used. It is based on the extraction of fat from milk and dairy products under the influence of concentrated sulfuric acid and isoamyl alcohol, followed by centrifugation and measurement of its volume using a graduated part of the lactobutyrometer. Before the treatment, the initial fat content of milk was determined by the standard procedure. Then, the milk was subjected to multiple treatments in a static cavitation device (Figure 1), described in detail in [28,29,54], and was analyzed at certain time points.

To estimate the homogenization quality of the milk, the measurements of the fat phase sizes were performed by means of a microscope, along with microphotography and computer analysis of the obtained data (analysis of the milk assay microphotography). The data were taken by means of a Micromed XS-2610 optical microscope (Ningbo Zhanjing Optical Instruments Co. Ltd., Yuyao city, China), with a web-camera attached, which in turn was connected to a PC. The preparation of the milk assays was as follows: assay mixing; selection and solution of the assay with distilled water in 1 to 40 proportion (three solutions were prepared from each assay and two compounds from each solution); spreading onto polished glass; keeping the assay for 20–30 min at room temperature; taking a photograph. The magnification while taking the photo of 10.4 × 10^−2^ m × 8.0 × 10^−2^ m size was 640 times (the camera resolution was 640 × 480 pixels). The number of fat globules and their diameters were calculated by means of the microscope. The calculation was performed taking advantage of the IBAS-2000 v.1.0 software. It enables the determination and analysis of more than 20 parameters of every fat globule: image projection surface, maximum and minimum diameters, and diameter similar to that of circle cross-section, in particular, as well as different parameters of image field, total number of globules, percentage of the field filled by globules, etc. The interval size of the fat globule groups was assumed to be 0.5 × 10^−6^ m. The number of intervals n was calculated approximately according to the formula $n=\sqrt{N}$, where N is the calculated number of fat globules. The mean diameter of the fat globules, mean quadratic deviation and coefficient of variation were determined.

### 2.3. Preparation of Fat Emulsions

In the laboratory, emulsions with a fat phase concentration of 35% were used. Reconstituted skimmed milk powder and plain milk fat and milk fat substitutes as a fat phase were used for its preparation, as follows: “Olmix 100 AK”, produced by the Kyiv Margarine Plant, “Fettimilk 02AK”, produced by Zaporizhzhya Fat-and-Food Complex, and “Delikon ZTL No. 1”, produced by Modified Fats Plant in Kirovohrad (Kropyvnytskyi). Emulsifiers: distilled monoglycerides and soy lecithin (in a ratio of 3:1) were used in the studies, which were added to the emulsion fat phase at 65° and 50 °C, respectively. The emulsions were evaluated according to their degree of destabilization, resistance and dispersion of the fat phase. Plain cream of the appropriate fat content was assumed to be a reference standard.

At this stage, the emulsification of the milk and oil fats in the rotor-type apparatus, as well as the effect of the main technological and energy parameters, was investigated (dispersion duration and intensity of the mechanical effect, mass portion of surface-active substances and fat phase in the emulsions, compositions and properties of the fat system, temperature).

### 2.4. Product and Methods for the Investigation of the Rheological Parameters of the Cream Cheese Mass

The main components of the product were cream cheese, water and flavors, with the content of the jelly substances being 1–5% of the total mixture mass. The content of cheese mass was 70.1%, water 17.7%, and flavors 12.2%, with a unified rheological coefficient of 3.193. At this stage, the effect of the rotor revolution frequency was investigated. The value of the shear stresses is dependent on the dispersion efficiency and the quality of the product structure, as well as the energy consumption during the thermomechanical treatment.

### 2.5. Temperature Measurement

The contact temperature was measured using the C-K thermocouples −50–0–800 °C, connected with the digital image potentiometer. While investigating the temperature in the flow of the operating mixture, the thermocouples were correspondingly technologically set. The engineering parameters of the C-K termocouples are those provided in [57].

### 2.6. Vacuum Measurement

Vacuum measurement during the mixture treatment was performed using the vacuum manometer, with the measuring limit of excess pressure being 5 × 10^5^ Pa of 1 class accuracy [58]. The vacuum manometer was connected to the operating camera of the test installation through the threaded hole in the operating camera cap.

### 2.7. Analysis of Microstructure

The investigation of the microstructure of the food products was performed on the basis of multiple magnifications of the prepared assay using optical devices; pictures were taken by means of a camera. In the considered case, the microstructural analysis of cream cheese masses was performed using a Motic light optical microscope (Motic Xiamen, Xiamen, China) with an integrated photo-video camera. During the experiment, the product assay was thinly spread on the polished glass using a microbiological loop. Afterwards, it was dried, covered by a glass cap and placed under the microscope eyepiece at 400× magnification.

### 2.8. Investigation of the Rheological Characteristics

The rheological characteristics were studied using the “RheoTest 2” (Rheotest Medingen GmbH, Ottendorf-Okrilla, Germany) stand. Yield viscosity was from 10^−2^ to 10^4^ Pa s; shear rate was from 0.1667 to 1458 s^−1^; shear stress ranged from 12 to 3000 Pa; temperature was from −30 to +150 °C. The measurement error was ±3% (for the Newtonian liquid). A new portion (assay) of the product was taken for each experiment. The data were recorded after the full rotation of the inner cylinder, with the product thermostatics given. Taking into account the structural–mechanical properties of the investigated product, the rheostat operation mode was where the “RheoTest 2” engine speed was 1500 rpm.

The prepared assay for the cream cheese mass was placed in the rheostat outer cylinder. Then, the sinking of the inner cylinder took place. The measurement of the tangential shear stress (Pa) was performed at twelve values of the shear rate gradient within a range of 0.33–145.8 s^−1^, the measuring scale data for device α were recorded at the maximum angular deflection of the device scale pointer.

### 2.9. Analysis of Rheological Characteristics

When analyzing the rheological characteristics of non-Newtonian liquids, the liquid-like systems without shear yield stress (the equations of Ostwald-de-Waele, Eyring–Powell, Barteneva–Yermylova, etc.) and those with shear yield stress (the equations of Shvedov–Bingam, Herschel–Bulkley, Kesson, Shulman, Myxajlov–Lyxtgejm, etc.) were determined. To describe the rheological characteristics of the cream cheese mass type food products, the equations of Ostwald-de-Waele and Herschel–Bulkley were used.

The shear stress (Pa) was calculated according to the formula:(1)$$\tau_{r}=Z\times \alpha$$
where Z is the cylinder constant, scale Pa/unit; α denotes the measuring device scale factors.

The effective viscosity (Pa · s) was calculated by means of the formula:(2)$$\eta_{eff}=\frac{\tau_{r}}{\gamma }$$
where γ is the shear rate gradient, s^−1^.

On the basis of the measurement results, the mathematic mean values were found, on the basis of which the graphs were plotted and further mathematic processing was carried out. The value of the shear yield stress and effective viscosity at the unit value of the rate gradient were determined using the methods of mathematic analysis and MS Excel software.

Dependence of the shear stress on the shear rate for the cream cheese mass can be presented as the Ostwald-de-Waele’s dependence:(3)$$\tau_{r}=B*\times \gamma^{m}$$
where $\tau_{r}$ is the shear stress between the product layers, Pa; B* is the viscosity at the unit value of the rate gradient, Pa·s; γ is the relative rate gradient that is numerically equal to the shear rate, s^−1^; m is the flow index.

To identify the influence of technological and mechanical factors on the rheological characteristics of the “Yagidka” dessert products, experiments were carried out in three stages: (1) under minimal mechanical processing and varying the temperature of product heating; (2) under constant temperature and varying the power of mechanical processing; (3) under the nominal apparatus mode. Only the average values of the studied parameters were analyzed. The values of effective viscosity at a single value of the velocity gradient, as well as the boundary shear stress, were found by mathematical analysis methods using the OpenOffice software suite.

## 3. Results

### 3.1. Milk Fat Homogenization Features in Milk

The experimental results regarding the change in the coefficient of milk homogenization depending on the treatment cycles number in the static cavitation device (Figure 2) at various temperature values versus the distance between the nozzle are shown in Figure 3. It was established that at six cycles of treatment, the best results for homogenization—78%—were obtained at a temperature of 72 °C. The mode of the developed cavitation was shown to be more optimal, being characterized by quickly providing the greatest number of vapor–gaseous bubbles in the shank of the cavitation cavity and the DIES effects presented in Table 1, as well as those described above. This mode can be achieved at the cavitation coefficient λ = 2.0–2.8. The cavitation coefficient specifies the intensity of the effect on the medium, together with the number of cavitations [59]. This coefficient was found as the relation of the vacuum cavity length to the obstacle diameter [54]. This criterion specifies the type of cavitation. Thus, when this coefficient is λ < 1, bubble cavitation is formed, at 1.5 < λ < 2.8, it is a developed cavitation, while at λ = 5 and more it is a mixed cavitation.

The greater efficiency of treatment at a temperature of 72 °C can be explained by a decrease in the viscosity of fat globules, the softening of their lipid membranes, and also by the fact that there is a partial thermal deaeration of the product.

During the research, it was found that the most effective number of cycles of milk processing with a fat content of 4% was six times. The homogenization efficiency increases along with increasing processing temperature. At a temperature of 20 °C it is 65%, at 50 °C, it is 70%, at 72 °C, it is 80%. In this case, this is a result of a decrease in the viscosity of fat globules, the softening of their lipid membranes and the melting of even the most refractory fractions of milk fat. This creates a “liquid–liquid” system, where partial thermal deaeration of the product takes place, in which the effectiveness of the hydromechanical influence of cavitation increases. Figure 3 shows the dependence of the milk homogenization coefficient on the distance between the narrowed area (L) and the obstacle in the six-fold treatment in a flow-type apparatus, with a milk temperature of 72 °C. The curve of the coefficient of homogenization versus the distance between the nozzle and the obstacle shows that there is an optimal distance from the nozzle to the obstacle, at which point the degree of homogenization is the highest. At a distance of 20 × 10^−3^ m following six-fold milk processing in the cavitation apparatus, the value of the coefficient of homogenization is within the range of 50–70%. According to the obtained plot, when increasing the distance between the nozzle and the obstacle to 60 mm, while maintaining a constant temperature and fluid speed, the homogenization rate decreases by 30–42%. With a further increase in distance, the homogenization increases to its maximum value of 78%, which is observed when there is a distance of 75 mm between the nozzle and the obstacle. When L changes, the cavitation coefficient increases from 2.0 to 2.8. The greatest number of the vapor–gaseous bubbles in the shank of cavitation cavity was observed at λ = 2.5 [54], at L = 75 mm.

It should be noted that further increase in the processing temperature did not result in improved process efficiency (data not presented in Figure 3), which could be a result of the fact that the intensity of the cavitation effect increases at relatively low temperatures, leading to a decreased elasticity of water vapor. In this case, the influence of cavitation occurs in a medium consisting of a liquid with solid fat globules distributed in it, that is, in a “liquid–solid body” system. The obtained effect is a result of the turbulent fluctuations during the impact of spherical waves and the accumulation of spray when cavitation bubbles are crushed. The mechanism of the homogenization effect is presented in Table 1.

The changes in the fractional composition of the fat globules according to their size after homogenization in the cavitation device, as compared to after valve homogenization, are presented in Figure 4, while microphotography of the milk assays is presented in Figure 5.

The milk parameters before homogenization were as follows: the mean diameter of fat globules d_mean_ = 2.6 × 10^−6^ m, dispersion $\sigma_{g}$ = 1.44, coefficient of variation (the portion of the feature scattering relatively the mean one) V = 67%. After valve homogenization and the corresponding impulse homogenization, these data were as follows: d_mean_ = 0.96 × 10^−6^ m and 0.9 × 10^−6^ m, $\sigma_{g}$ = 0.50 and 0.46, V = 51 and 56%. The mean diameter of the fat globules during treatment in the cavitation device decreased by 6%, compared to the valve one. Additionally, the dispersion value also decreased, which, in its turn, indicates that the homogenization in the cavitation device results in the stability of the milk fat phase after homogenization.

According to the data presented in Table 1, dispersion takes place as a result of the discrete-impulse energy supply when turbulence (theory of isotropic turbulence [39,60]) is not the only the active factor in the static-type apparatus (Figure 1), but also the effects related to the life cycle of the cavitation bubbles. In the paper by J. Hinze [11], an analysis of droplet deformation and crushing was carried out from the point of view of the turbulence theory by A. Kolmogorov, whereby the stochastic nature of the phenomenon was specified and its connection with different types of flow was demonstrated. The criteria by Weber and Laplace presented in this paper were treated as the terms of the isotropic turbulence theory, and it could not be proved that the turbulence pulsations themselves, rather than the flow macroparameters, were the cause of droplet crushing. It should be noted that the experiments, on the basis of which the empirical relations evidencing the efficiency of droplet crushing as a result of turbulence flow parameters, were carried out according to the “black box” principle [61]. Under the resulting turbulence flow parameters, the distribution of the secondary droplets was fixed according to size, or the value d_max_ was found. However, the crushing itself is not within the scope of our investigations. This is why many investigators are not sure about the application of the theory of turbulence being the only relevant concept [62,63,64]. However, it can be assumed that turbulence makes a certain contribution to the dispersion processes. The other significant factor is the cavitation component, which to a great degree determines the dispersion efficiency, because the aggregation of bubbles, developing dynamically, creates the fields of pressure and velocities in the liquid that are very similar to the structure of turbulence flow.

One significant difference is that the bubble mass is the place where extremely high amplitudes of pressure impulse and velocity are initiated. If the bubbles are being buffeted close to the dispersion particles, the kinetic energy of the liquid radial movement is transformed into the mechanical energy of the cumulative microspray flying out from the bubbles at extremely high velocity towards the dispersion particles. It should be noted that the cumulative mechanism proves the principle of the discrete impulse energy supply: energy localization within a short time and limited spatial area, as well as the directed impact effect of concentrated energy as the impulse.

The effect of the distance between the nozzle and the obstacle on the homogenization rate can be explained in the following way. When changing the position of the obstacle relative to the nozzle, the geometry and volume of gas cavities change as well. The most active formation of small-sized caverns, and their rapid collapse just after an obstacle, without reaching the next level, was observed at distances of 20 and 100 mm. With increasing distances, the size of the cavity increases, and the intensity of their formation decreases, resulting in a decrease in the homogenization coefficient.

These results are confirmed by the data given in [24,25,26,27]. The authors note that the processing of milk and cream in cavitation devices affects the size of fat globules; specifically, the diameter of the fat globules decreases by up to 1.0–1.5 × 10^−6^ m, increasing the relative viscosity of the product. In [24], the effect of hydrodynamic cavitation on the preparation of whole milk substitutes with fat globules with a size of 0.5–2 × 10^−6^ m was studied. The efficiency of cavitation treatment is a result of the high rate of flow in the fluid processes that determine the mechanical model of cavitation action. These include: the creation of intense fields of pulsating pressures and waves of discharge; the formation of cumulative micro streams; the formation of turbulent zones in the flow just after the obstacle with swirling and collapsing microbubbles; phase transitions occurring on the surface of cavitation bubbles; and temperature fluctuations in the vapor–gas content of the cavities at their collapse.

The results of the organoleptic estimation indicated that the milk patterns, in terms of their taste, smell, consistency and appearance, varied depending on their mode of treatment. At λ = 1.0–1.5, the milk taste was clearly sweet, and its consistency, appearance and smell were no different from those of the initial plain milk. Treatment at λ = 2.0–2.8 caused the formation of a light, grain-like consistency and the smell of pasteurized milk, with an appearance that was similar to that of the control pattern. At λ = 2.8–5.0, the milk consistency was grain-like and non-homogenous, and the milk took on the nonspecific bitter taste of burnt milk, staying milky-white in color. It should be noted that investigations are being undertaken with the aim of studying the effects of different milk treatment modes under DIES. The main purpose is still to increase milk quality while completely preserving its value.

### 3.2. Patterns of Fat Dispersions Formation under Discrete-Impulse Energy Supply

The Couette flow and the shear stresses in the dispersion processes were mentioned above. In [15], the main concepts behind the mathematic model of the deformation and crushing of droplets in liquid and gaseous states, based on the principles of shear and accelerated flows, specifying the rotor-pulsation apparatuses, were presented.

The results obtained for the formation patterns of fat dispersion in a rotary device indicate that the process of fat emulsification in milk plasma is cyclical in its nature (Figure 6 and Figure 7). After obtaining a dispersion with an average size of fat globules (<4 × 10^−6^ m), the process of dispersion and the process of aggregation of fat globules alternate, accompanied by increases and decreases in the stability of the emulsion. The most effective emulsification takes place at a rotor speed of the dispersing device of 3000 rpm (Figure 7). In this case, after 1.5 min of treatment, a finely dispersed (fat globules with average size <2 × 10^−6^ m) and durable emulsion (degree of destabilization of about 30%) was formed. The emulsion, similar to that with the described characteristics, was also obtained at a rotation speed of 2500 rpm. Increasing the emulsion treatment intensity in the rotary device results in an increase in power. Therefore, a speed of 1500 rpm corresponds to about 60 Watts, while 3000 rpm corresponds to 150–200 Watts.

It was determined that in order to obtain an emulsion with dispersion and stability qualities similar to those of plain creams, it is preferable to perform emulsification without vacuuming at a temperature of 50–70 °C. As the surfactant concentration decreases, the optimal treatment duration also decreases. However, excessive emulsifier does not contribute to the stability of the emulsion. In this case, the optimal concentration of surfactant is 0.6% (distilled monoglyceride + lecithin in the ratio of 3:1), which ensures the production of a technologically stable emulsion of 35% fat, using a rotary-type device (Figure 8).

These results demonstrate that the use of the fat phase of different types of fats does not significantly affect the emulsification process (Figure 9); moreover, it has little effect on power or energy consumption during emulsification. At the same time, increasing the concentration of the fat phase retards the formation of the emulsion, as it increases its degree of destabilization and, accordingly, the optimal duration of emulsification. This, as well as the increased viscosity of the emulsion along with its fat content, leads to an increase in the power and energy cost of emulsification by ≈1.6 times, which increases with the fat content when it is increased from 3.5 to 35%.

Taking everything into account, we can state that the optimum parameters for obtaining an emulsion with a fat content of 35% (with dispersion and stability similar to those of plain creams) in a rotary-type device, regardless of the type of fat phase used, at a surfactant concentration of 0.6%, are the following: an emulsification temperature of 50–70 °C, a processing power of 150–200 Watts, an emulsification device rotor speed of 3000 rpm, and a processing time of 1–1.25 min; the energy consumption in this case is 2–3.5 kJ/kg.

### 3.3. The Rheological Characteristics Variations of Cream Cheese Masses

In addition to the dispersion efficiency and resistance of the complex products of milk processing, the rheological characteristics and the product structure are of great importance. The dependence of the viscosity of the dessert cheese mass on the shear rate for the temperature range under study evidences the increase in the shear rate and the decrease in the effective viscosity (Figure 10 and Figure 11).

With an increase in temperature from 10 °C to 30 °C, the viscosity and shear rate decrease, and remain approximately constant within the temperature range of 30–50 °C, whereby they increase with increasing temperature. This happens due to the structure of the thickener (pectin), for which the temperature of gelling is in the range of 52–55 °C. The boundary shear rate makes it possible to take into account the dependence of Herschel–Bulkley.
(4)$$\tau_{r}=\tau_{0}+B\times \gamma^{m}$$
where $\tau_{0}$ is the boundary shear rate, Pa.

The pattern of influence of the heating temperature on the effective viscosity is maintained. The value of the structure destruction rate, which varies within fairly small limits, confirms the validity of the experiment. For an intact structure at an initial temperature of 10 °C, the dependence assumes the following form:(5)$$\tau_{r}=9.1+11.5\times \gamma^{0.475}$$

At 65 °C (at the end of the heating process), the dependence is as follows:(6)$$\tau_{r}=9.3+9.6\times \gamma^{0.429}$$

When cooled 50 °C it looks like:(7)$$\tau_{r}=9.6+10.0\times \gamma^{0.434}$$

The dependence graphs of the boundary shear rate and effective viscosity on the duration of mechanical treatment at a temperature of 10 °C are shown in Figure 12 and Figure 13.

Under the condition of no mechanical processing, the boundary shear rate increases with the increasing angular deformation rate. At the same time, the destruction of the structure is observed in the first cycles of processing. Hereafter, the boundary shear rate, as well as the other rheological characteristics, remain almost constant. However, when the duration of processing of the cream cheese mass exceeds 260 cycles of mechanical treatment, an increase in the boundary shear rate is observed, and the effective viscosity increases slightly. This happens due to the processes taking place at the micro level, the phenomena of cavitation, local heating, and the swelling of the structure-forming agent in particular.

The Ostwald-de-Waele rheological equation for the mass at the initial stage of the process is as follows:(8)$$\tau_{r}=12.5\times \gamma^{0.472}$$

The rheological equations at the end of the process in the form of the Ostwald-de-Waele dependence assume the following form:(9)$$\tau_{r}=5.1\times \gamma^{0.195}$$

In the Herschel–Bulkley dependence:(10)$$\tau_{r}=5.5+1.16\times \gamma^{0.370}$$

It should be noted that the destruction of the mass structure takes place during the first 20 cycles of product circulation, and the mass heating takes place in a completely destroyed structure. On the basis of the analysis of the product movement process in the rotor–stator pair, it is evident that not all of the transported product, but only part of it (Figure 14), is subjected to mechanical grinding.

One can distinguish the zones in which the nature of the mass movement will be different: zone I is the area of the most active mechanical effect on the product (the width of this zone at the beginning in the direction of product movement $d_{c}$ is similar to that the size of the transported mass globules with adjacent layers), in which the grinding between a rotor and a stator occurs at a large velocity gradient and at low stress on a cut of cream cheese masses; in zone II, the transportation of the product mainly occurs without its being mechanically processed, with the exception of a degree of mass exchange between zone I and II. The product that is captured by the grooves will also be destroyed.

Let us denote the volume of the loaded mass by $V_{0}$. Then, the volume of the mechanically treated product after one cycle of passing through a rotor–stator pair will be $\psi \cdot V_{0}$. Therefore, the volume of the unprocessed product will be $\left(1-\psi \right)\cdot V_{0}$.

Due to the peculiarities of the geometric parameters of the mass flow channels, it was assumed that the coefficient of proportionality ψ corresponds to the ratio of the total area of zone I to the total area of zones I and II in the lower part of the rotor–stator pair.

Let us analyze several stages of the mechanical processing of the product during its movement in the circulation circuit of the apparatus; the results are presented in Table 2.

Let us denote the running volume of the treated material by $V_{b}$, the product circulation cycle in the rotor–stator pair by i, the required number of cycles of mechanical treatment by r and the number of cycles of mechanical treatment at this stage by κ.

Then, the dependence of the number of cycles of processing on running volume will look like:(11)$$V_{b}=\Delta \psi^{\kappa }\times {\left(1-\psi \right)}^{i-\kappa }\times V_{0}$$
where Δ is the numerical coefficient.

In the general case, it assumes the following form:(12)$$\Delta =\Delta \left(\begin{array}{c} \kappa \\ i \end{array}\right)=\frac{i!}{\left(i-\kappa \right)!\kappa !}$$

Then, Formula (9) can be presented as follows:(13)$$V_{b}=\frac{i!}{\left(i-\kappa \right)!\kappa !}\times \psi^{\kappa }\times {\left(1-\psi \right)}^{i-\kappa }\times V_{0}$$

Using the dependence (10), it is possible to estimate the level of mechanical processing of the loaded product. For example, in order to obtain the relevant technological qualitative indicators, it is necessary to ensure the number of cycles of mechanical treatment r. Then, for the quantitative estimation of mechanical processing at an arbitrary stage of processing of raw materials $V_{r}$, it is necessary to find the sum of all of the parts of the raw materials other than those that have been processed fewer than r times:(14)$$V_{r}=V_{0}\overset{i}{\underset{\kappa =r}{\sum }}\frac{i!}{\left(i-\kappa \right)!\kappa !}\times \psi^{\kappa }\times {\left(1-\psi \right)}^{i-\kappa }$$

It is proposed that the ratio of the sufficiently treated raw material $V_{r}$ to its initial quantity $V_{0}$ be considered as an index (coefficient) of mechanical treatment in the cyclical apparatus:(15)$$\xi =\frac{V_{r}}{V_{0}}=\frac{V_{0}\sum_{\kappa =r}^{i}\frac{i!}{\left(i-\kappa \right)!\kappa !}\times \psi^{\kappa }\times {\left(1-\psi \right)}^{i-\kappa }}{V_{0}}=\overset{i}{\underset{\kappa =r}{\sum }}\frac{i!}{\left(i-\kappa \right)!\kappa !}\times \psi^{\kappa }\times {\left(1-\psi \right)}^{i-\kappa }$$

Let us assume that ten-fold mechanical processing is sufficient for proper maintenance of the finished product structure. From the dependence (Figure 15c), according to the Formula (10), it is evident that a guaranteed ten-fold processing of the product will be provided at 45 cycles of circulation of the product through a rotor–stator pair. The microstructure analysis of the “Yagidka” product, performed by means of a Motic optical microscope with an integrated photo-video camera (Figure 15a,b) at the initial moment and at the time corresponding to 45 cycles of mechanical processing, proves that this duration of mechanical processing is sufficient to ensure the formation of the structure of the product and provide an equal distribution of its components in the volume. The maximum particle size was 0.2 mm. Thus, it can be noted that the required level of mechanical treatment was achieved long before the completion of the process cycle.

## 4. Conclusions

For the qualitative estimation of the discrete-impulse energy supply in the system specified by the accelerated movement of the continuous phase and the turbulence, cavitation mechanism and the shear stresses provided in the static-type and rotor-pulsation cavitation apparatus, a series of investigations pertaining to the milk homogenization peculiarities, emulsification of milk and oil fats and the dispersion of the viscous pseudo-plastic object with the structure-forming components was carried out.

It was determined that the application of hydrodynamic cavitation as an active component in discrete-impulse energy supply mechanisms provides an active dynamic effect with respect to the dispersion particles of the emulsion systems:To obtain a milk emulsion dispersion equal to 0.8 m^−6^, six cycles of processing must take place under the mode of the developed cavitation specified by the great number of vapor-gaseous bubbles, the cavitation coefficient being λ = 2.5;While increasing temperature from 20 °C to 72 °C, the content of homogenized milk fat increased from 60% to 80% as a result of a decrease in the viscosity of the fat globules, the softening of their lipid layer, and the melting of the most refractory fractions of the milk fat. With an increase in temperature to 72 °C, partial thermal deaeration of the product occurred, as a result of which cavitation bubble initiation processes became easier, their number became greater, and, as a result, the number of local microreactors of local or discrete transformation increased as well.

The investigations pertaining to the emulsification of milk and oil fats in the rotor-pulsation apparatus proved that:Having obtained a dispersion of less than 4 × 10^−6^ m with medium-sized fat globules, their dissipation and aggregation subsequently took place; over a dispersion period of 60–75 s, with a surface-active substance content of 0.6%, a stable fat emulsion was achieved;When the mass portion of the fat phase was increased from 3.5 to 35%, the duration of the process increased as a result of the greater degree of emulsion destabilization. At the same time, when the fat comprises different kinds of fat, there is an insufficient effect on the emulsification, the dispersion efficiency, or the duration of processing;No clear dependence was found between the revolution velocity of the emulsification apparatus rotor and the dispersion of the emulsion fat phase, although some confusion was caused by the emulsion dispersion result obtained when the processing velocity was 3000 rpm. The average sizes of the fat globules were greater than those in emulsions obtained under slower revolutions. Additionally, the data obtained proved that while being processed in the emulsification apparatus (with a rotor velocity of 1500–3000 rpm), a finely dispersed fat emulsion with globule sizes commensurable with those of the plain milk cream could be obtained in 90 s.The conventional parameters required to achieve technological stability for the production of fat emulsion spreads with dispersion and stability commensurable to those of plain milk creams were: emulsification temperature 50–70 °C, surface-active substance content 0.6% without vacuuming, treatment power 455–220 Vt, emulsion device rotor velocity 3000 rpm (18.8 m/s); treatment duration 60–75 s, power expenditure 1.9–3.3 kJ/kg.

According to the results of the investigation related to the cream cheese mass dispersion efficiency, it should be stated that:The crushing of the complex protein product structure occurs during the first 20 cycles of product processing. Mass heating was performed by the heat supplied from outside and by the mechanical work in the rotor–stator couple;The rheological characteristics of the product under the action of mechanical and thermal factors were presented on the basis of Ostwald-de-Waele’s and Herschel–Bulkley’s equations;It was proposed that the degree of mechanical treatment of the product be determined by the factor, that is, the relation of the sufficiently treated mass volume to the general mass volume;An equation for the assessment of the degree of mechanical treatment of any part of the product was proposed.

## Figures and Tables

**Figure 1 materials-14-04181-f001:**
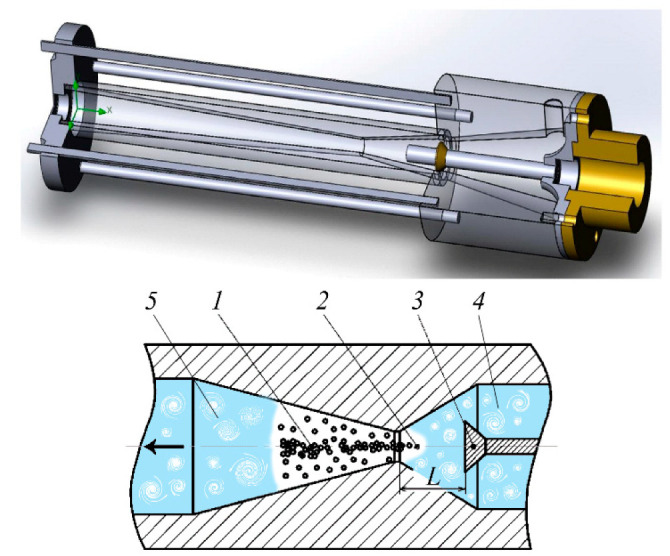
Static cavitation device: 1—active processing area (cavitation flow); 2—narrowed area; 3—cone; 4,5—turbulent flow; L—the distance between the nozzle and the obstacle.

**Figure 2 materials-14-04181-f002:**
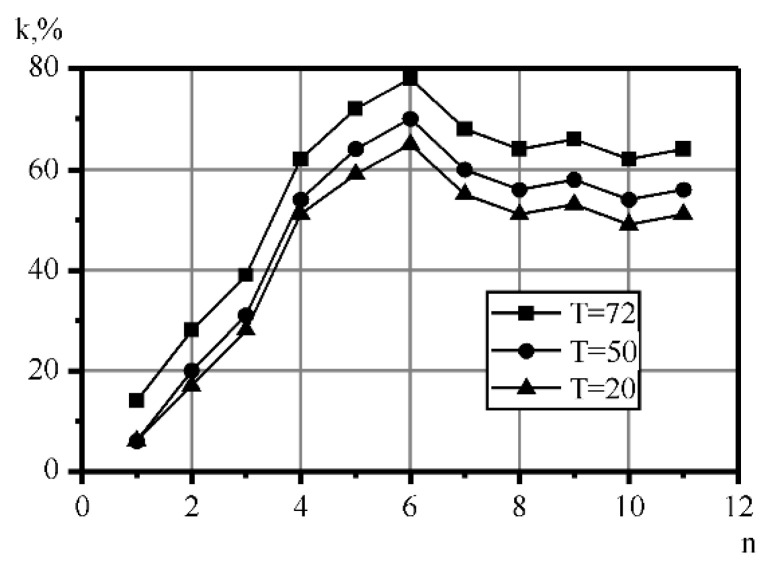
Milk homogenization coefficient versus the number of processing cycles at different temperature values curve (cavitation coefficient λ = 2.5): 1—20 °C, 2—50 °C, 3—72 °C.

**Figure 3 materials-14-04181-f003:**
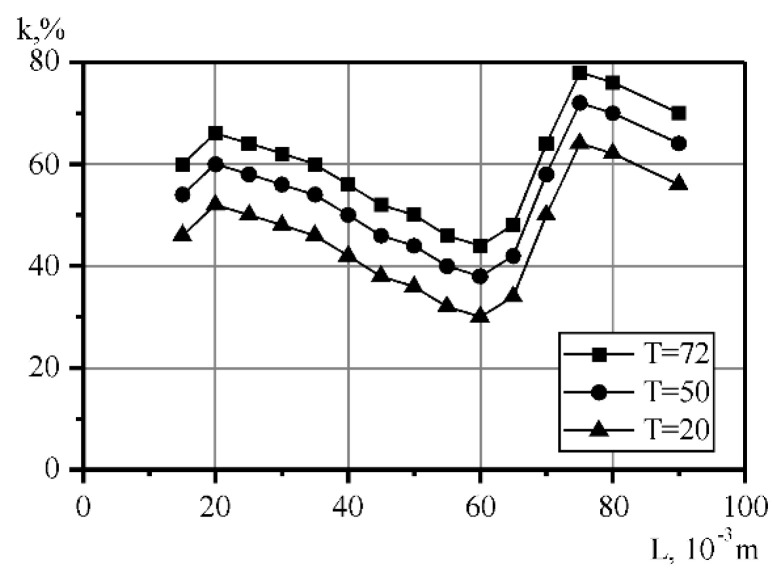
Milk homogenization coefficient versus the distance between the nozzle and the obstacle curve, when *n* = 6: 1—20 °C, 2—50 °C, 3—72 °C.

**Figure 4 materials-14-04181-f004:**
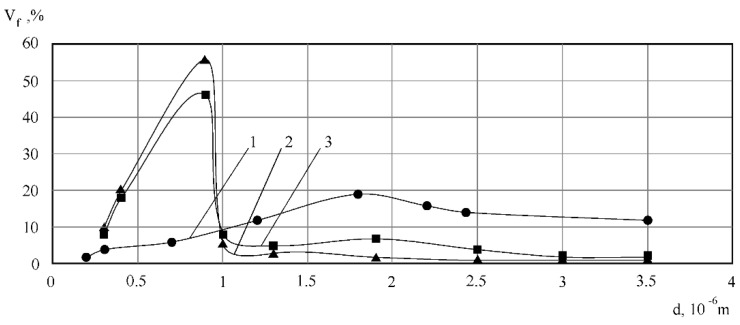
Volume distribution of the fat globules according to their sizes: 1—non-homogenized milk; 2—milk after the valve homogenization; 3—milk after treatment in the cavitation device.

**Figure 5 materials-14-04181-f005:**
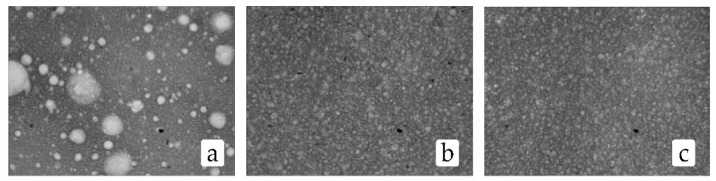
Microphotography of milk: (**a**) non-homogenized; (**b**) after the valve homogenization; (**c**) after treatment in the cavitation device.

**Figure 6 materials-14-04181-f006:**
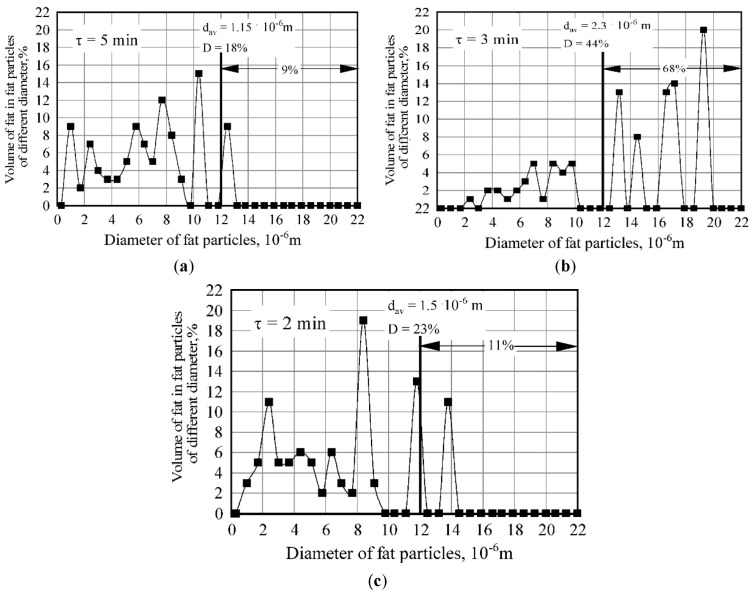
Variance in the granulometric composition of fat emulsion during processing in the emulsification device (*n* = 2500 rpm): (**a**) τ=5 min; (**b**) τ=3 min; (**c**) τ=2 min.

**Figure 7 materials-14-04181-f007:**
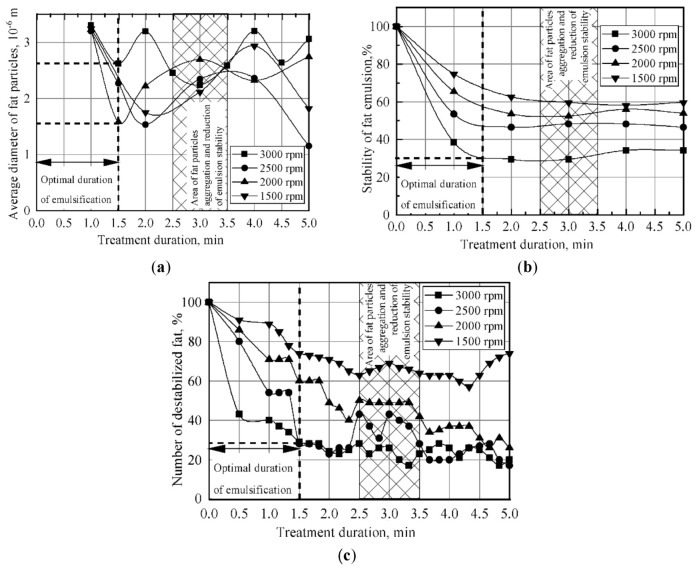
Variance in the degree of dispersion and the stability of fat emulsion in the process of treatment in the emulsification device: (**a**) average diameter of fat particles; (**b**) stability of fat emulsion; (**c**) number of destabilized fat.

**Figure 8 materials-14-04181-f008:**
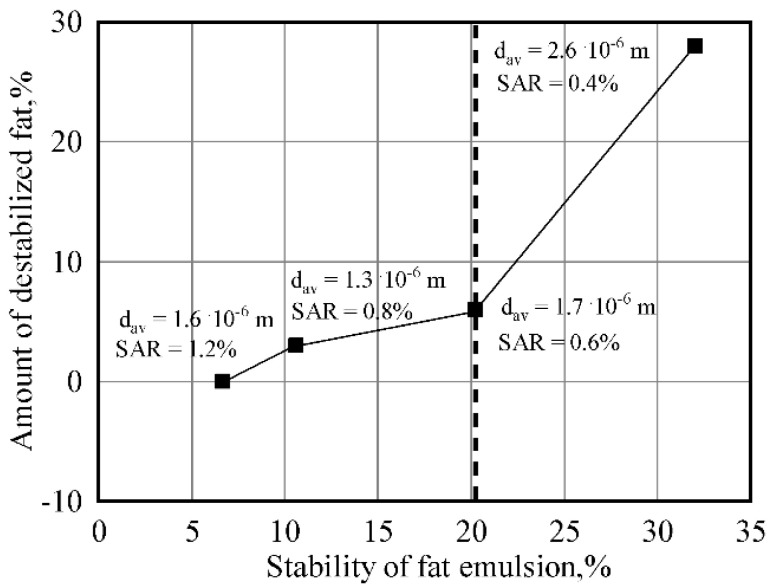
Variations in the degree of dispersion and fat emulsion stability as a function of surfactant concentration (concentration of surfactant, %: monoglyceride + lecithin in the ratio of 3:1; τ = 1.5 min).

**Figure 9 materials-14-04181-f009:**
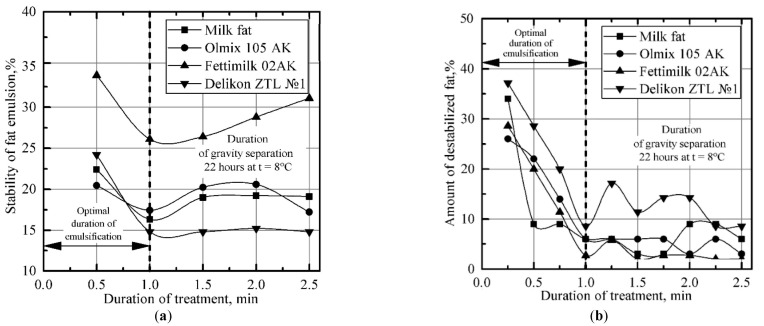
Variance in fat emulsion stability during treatment in the emulsification device (concentration of surfactant 0.6%: monoglycerid + lecithin in the ratio of 3:1): (**a**) stability of fat emulsion; (**b**) amount of destabilized fat.

**Figure 10 materials-14-04181-f010:**
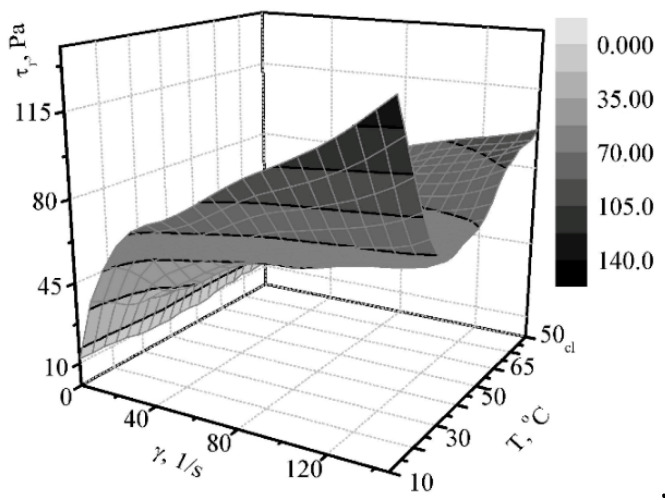
The dependence of the shear rate for the “Yagidka” product on shear rate and time with minimal mechanical treatment.

**Figure 11 materials-14-04181-f011:**
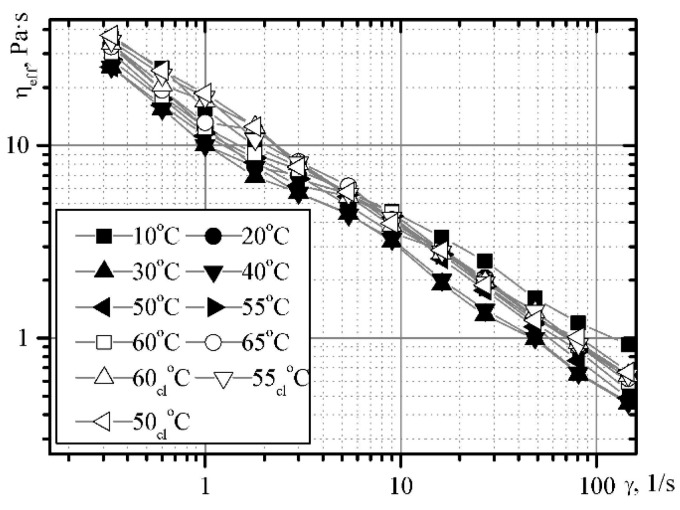
The dependence of the effective viscosity on the shear rate for the “Yagidka” product with minimal mechanical treatment.

**Figure 12 materials-14-04181-f012:**
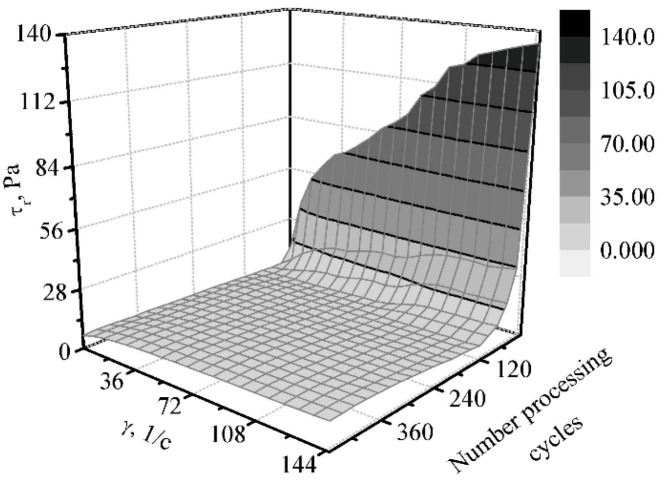
The dependence of the shear rate for the “Yagidka” product on the shear rate in the emulsifier at a temperature of 10 °C.

**Figure 13 materials-14-04181-f013:**
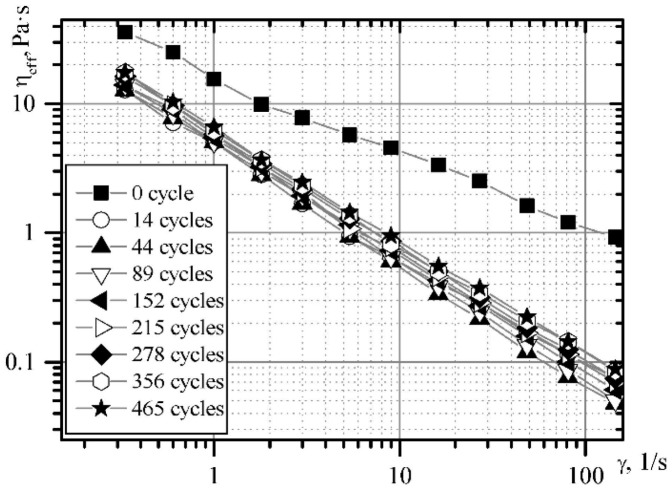
The dependence curve of the effective viscosity for the “Yagidka” product during the treatment in an emulsifier at a temperature of 10 °C.

**Figure 14 materials-14-04181-f014:**
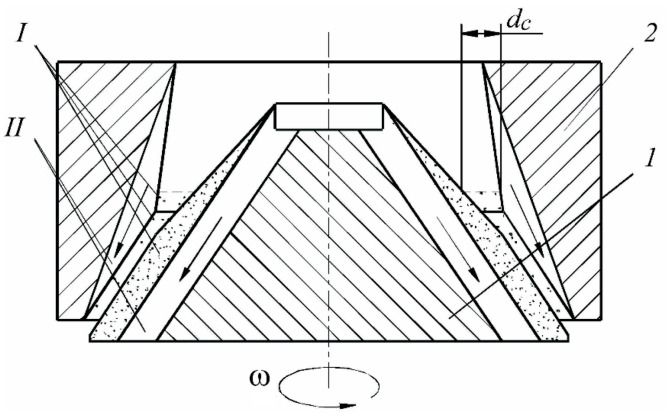
The scheme of movement of cream cheese mass in the channels of the rotary–vortex emulsor. 1—rotor, 2—stator; I—zone of transportation and mechanical treatment of cream cheese mass; II—zone of transportation without mechanical processing of cream cheese mass.

**Figure 15 materials-14-04181-f015:**
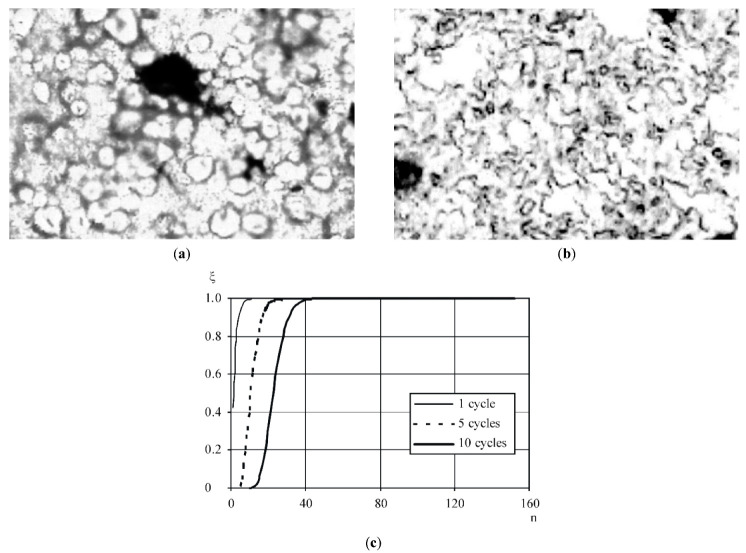
Microstructure of the “Yagidka” cream cheese mass: (**a**) the microstructure of the cream cheese mass before treatment; (**b**) the microstructure of the cream cheese mass after 45 cycles of treatment; (**c**) the level of mechanical treatment of the product as a function of the number of processing cycles.

**Table 1 materials-14-04181-t001:** Mechanism of milk fat dispersion.

Turbulent drop dispersion: isotropic and viscous. The mechanism of isotropic dispersion turbulence is followed by the stress fluctuations caused by the microvorticity. Under the viscous mechanism it is caused by the shear stress of greater vorticity [60] (turbulent regime zone).Turbulence caused by the pulsation and buffeting of the vapor–gaseous cavitation bubbles.	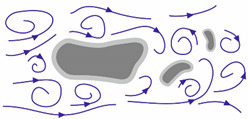
In the cavitation zone, fluctuation cavitation bubbles arise, which, colliding with the dispersion phase drops, are buffeting.Cumulation sprays formed in the bubbles strike the fat globule and break it into small ones. The fat globule pulls into the bubble and crashes into the dispersion phase fat globule.	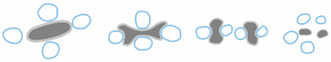
Fat phase dispersion caused by the high location pressure differential (impact waves) during the cavitation bubbles buffeting.I—cavitation bubble formation, II—bubble of the maximum size, III—sizes decrease, IV—bubble explosion followed by the cumulation spray.	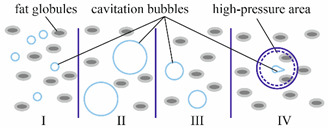

**Table 2 materials-14-04181-t002:** Mechanical processing of the product in the rotor–stator pair depending on its circulation.

Circulation Cycle	Mechanical Processing of the Product (Number of Cycles)			
	0	1	2	3
0	$V_{0}$			
1	$\left(1-\psi \right)\times V_{0}$	$\psi \times V_{0}$		
2	${\left(1-\psi \right)}^{2}\times V_{0}$	$2\times \psi \times \left(1-\psi \right)\times V_{0}$	$\psi^{2}\times V_{0}$	
3	${\left(1-\psi \right)}^{3}\times V_{0}$	$3\times \psi \times {\left(1-\psi \right)}^{2}\times V_{0}$	$3\times \psi^{2}\times \left(1-\psi \right)\times V_{0}$	$\psi^{3}\times V_{0}$

## Data Availability

The data presented in this study are available on request from the corresponding authors.

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
