# Peer review of "Discrete-Impulse Energy Supply in Milk and Dairy Product Processing"

_materials, 2021, doi:10.3390/ma14154181_

Round 1

Reviewer 1 Report

References of 45-51th is written with bold characters....

Author Response

Dear colleagues,

first of all we want to express our gratitude to the reviewers for comments on the article, which will help to present it better.

As for the the comments:

  1. According to the first comment we have made corrections to the abstract which allows to clarify the results of the work.

2.The introduction also includes adjustments to research in this area. Regarding the reduction of the volume of the introduction, we would like to note that previous reviewers noted that the coverage of the issue should be more detailed, so we added information on the description of the DIES mechanism and methods that allow to obtain the effects that accompany DIES. If we cut off this information it will be not clear how DIES, cavitation and hydromechanic processes of homogenisation and emulsification are combined

  1. Brief information about sensory changes in food obtained by processing is added too. The expanded presentation will add more volume.
  2. A number of changes to conclusions have also been made to present the results in a more concrete way.
  3. References have also been adjusted.

With my best regards,

Sincerely yours,

Ph.D., D.Sc. (Eng.) Paweł Droździel

Reviewer 2 Report

The authors approached an interesting topic with potential and a good scientific basis, however the paper needs to be structured and completed to make the paper easier to understand and more consistent with the experiments performed.

Remarks:

- the abstract is superficial and does not provide relevant results for the purpose of the paper.

- the introduction should make a more concise presentation of the processes and provide more details about the research results

- in presenting the results, I believe that some concrete references should be made to possible sensory changes in food obtained by processing.

- a clearer and more concise expression is preferably required

Author Response

(The authors gave the same response as above.)

Reviewer 3 Report

The authors introduced the nomenclature table as I had asked them (april 11, 2021), thus making clear the definition of the quantities present in the manuscript. In addition, the introduction, materials and methods, results and conclusions have been expanded, making the research work carried out clearer.

Author Response

(The authors gave the same response as above.)

Round 2

Reviewer 2 Report

Improvements have been made to the manuscript.

This manuscript is a resubmission of an earlier submission. The following is a list of the peer review reports and author responses from that submission.

Round 1

Reviewer 1 Report

This paper is dedicated to the study of the effect of DIES and other emulsification methods on the stability of various dairy products. The results presented in this paper are very interesting and of great interest for the dairy industry. 

I recommend accepting this paper after correcting grammatical and expression errors in the text.

Author Response

Dear colleagues,

We appreciate your comments and suggestions while revising our article. The article has been corrected due to recommendations and we are sending it again.

Make sure of the rows in the article being changed because of correction

Reviewer 1

According to the reviser’s remarks the article has been corrected.

Reviewer 2 Report

Row 183: Pa·c? → Pa·s

Row 183: 10-2? → 10-2

Row 183: c-1? → s-1

Row 193: s-1? → s-1

Row 199: Gerschel-Bulkli? → Herschel-Bulkley

Row 205: Viscisity? → Viscosity

Row 205: (Pa·sec)? → (Pa·s)

Row 206 and 216: s-1? → s-1

 Fig. 1 and Fig. 2: they appear reversed

Fig. 3 and other figures that follow: there are quantities which are not defined as for example M in fig. 3. A table with the nomenclature of all the quantities present in the manuscript is needed. The units of measurement adopted are also needed, for example what does the unit mkm mean for the diameter d?

Row 228, 229 and 230: it is not clear how the figures are cited

Fig. 2 and row 249-260: The authors talk about the distance between the nozzle and the buffle, but there is no clarifying scheme. They cite the scheme of one of their work, (Erosive ... in Diagnostika) but looking at the scheme in their cited work there is no explanation of the nozzle-buffle distance. So they should report the complete schematic here. The authors talk about the distance between the nozzle and the buffle, but there is no clarifying scheme. They cite the scheme of one of their work, (Erosive ... in Diagnostika) τττlooking at the scheme in their cited work there is no explanation of the nozzle-buffle distance. So they should report the complete schematic here.

Row 305: it would be useful to add the quote from figure 13, useful to understand immediately how the rotary device works.

Row 425: syntax of sentence?

Row 425 and 430: What is the difference between “multiplicity of processing af stage K” and “multiplicity of mechanical treatment r”? It needs a complete definition in a proposed nomenclature table.

Eq. (11): Where does it come from? authors must explain how they get it or cite a source.

Row 427: eq. (9)? But this provides τ.        

Fig. 14: What is ε, the level of mechanical treatment? It needs a complete definition in a proposed nomenclature table. The number of cycles is reported on the abscissa and then there are three parametric curves of the number of cycles, 1, 5 and 10. The authors must explain better that they are two different quantities and therefore it is necessary to define the relative symbol and the description in the table of the nomenclature.

Eq. (14): Vr - V0·(…) or Vr = V0·(…) ?

Author Response

Dear colleagues,

We appreciate your comments and suggestions while revising our article. The article has been corrected due to recommendations and we are sending it again.

Make sure of the rows in the article being changed because of correction

Reviewer 2

  1. The authors agree with your remarks Row 183: Pa·c? → Pa·s Row 183: 10-2? → 10 Row 183: c-1? → s Row 193: s-1? → s Row 199: Gerschel-Bulkli? → Herschel-Bulkley Row 205: Viscisity? → Viscosity Row 205: (Pa·sec)? → (Pa·s) Row 206 and 216: s-1? → s Fig. 1 and Fig. 2: they appear reversed.

The article was corrected according to the suggestions. (Row 183, 183, 183, 193, 198, 205, 206, 216). There are Fig. 2 and Fig. 3 according to a new redaction, and other remarks.

  1. 3 and other figures that follow: there are quantities which are not defined as for example M in fig. 3. A table with the nomenclature of all the quantities present in the manuscript is needed. The units of measurement adopted are also needed, for example what does the unit mkm mean for the diameter d?

We agree with your remarks. The Table with the nomenclature of all the quantities has been enclosed (table 2). The diameter is in meters according to the SI system (fig. 4).

  1. Row 228, 229 and 230: it is not clear how the figures are cited

According to your remark the notation L is introduced (Fig 1, row. 128, row. 245)

  1. 2 and row 249-260: The authors talk about the distance between the nozzle and the buffle, but there is no clarifying scheme. They cite the scheme of one of their work, (Erosive ... in Diagnostika) but looking at the scheme in their cited work there is no -2 -1 -1 -1 © 1996-2021 MDPI (Basel, Switzerland) unless otherwise stated Disclaimer Terms and Conditions (https://www.mdpi.com/about/terms-andconditions) Privacy Policy (https://www.mdpi.com/about/privacy) (g) g explanation of the nozzle-buffle distance. So they should report the complete schematic here. The authors talk about the distance between the nozzle and the buffle, but there is no clarifying scheme. They cite the scheme of one of their work, (Erosive ... in Diagnostika) τττlooking at the scheme in their cited work there is no explanation of the nozzle-buffle distance. So they should report the complete schematic here.

We agree with it. The scheme and the model of the device is added. (Fig. 1)

  1. Row 305: it would be useful to add the quote from figure 13, useful to understand immediately how the rotary device works.

We agree with your remark. Citation is added. (Row 305)

  1. Row 425: syntax of sentence?

The sentence has been corrected. (Row 436)

  1. Row 425 and 430: What is the difference between “multiplicity of processing af stage K” and “multiplicity of mechanical treatment r”? It needs a complete definition in a proposed nomenclature table.

We agree with your remark. The text (Row 402) was provided with these data (factors)

  1. (11): Where does it come from? authors must explain how they get it or cite a source.

We agree with it. The Table 3 (Row 400) is presented and the data from it are the source for the formula (11).

  1. Row 427: eq. (9)? But this provides τ.

The text has been corrected. (Row 371) (Eq (3), (Eq (9))

  1. 14: What is ε, the level of mechanical treatment? It needs a complete definition in a proposed nomenclature table. The number of cycles is reported on the abscissa and then there are three parametric curves of the number of cycles, 1, 5 and 10. The authors must explain better that they are two different quantities and therefore it is necessary to define the relative symbol and the description in the table of the nomenclature.

We agree with it and the Fig. 15c has been corrected.

  1. (14): V - V ·(…) or V = V ·(…) ?

The text is corrected. (Row 416)

  1. The proper English revision of terms has been done.

Reviewer 3 Report

The title and research work does not fit well. The abstract is poorly written. Extensive English revision is required. Need clear information of types of emulsion and dispersed phase compositions. Need lots of improvement on the data representation and statistical analysis, since the data is scattered.   

Author Response

Dear colleagues,

We appreciate your comments and suggestions while revising our article. The article has been corrected due to recommendations and we are sending it again.

Make sure of the rows in the article being changed because of correction

Reviewer 3

  1. The title and research work does not fit well.

We think the title of our article is well. But we can offer another redaction of the title: “Regularities of the hydro-mechanical processes for the milk and dairy products processing under the discrete-impulse energy supply”.

  1. The abstract is poorly written.

The abstract is written properly according to your suggestion.

  1. Extensive English revision is required.

Proper English revision has been done.

  1. Need clear information of types of emulsion and dispersed phase compositions.

Only the main characteristics of the investigated emulsions have been presented due to a great amount of investigations to be carried out. Sufficient information will be available in the monography being prepared now. 

  1. Need lots of improvement on the data representation and statistical analysis, since the data is scattered.

The Table 2 on the fat globule sizes has been provided in the article. The other statistic data are being generalized and are not available in the article because of the article limits.

Reviewer 4 Report

The topic of the study is interesting and can bring important clarifications to the processes. 

I believe that references can be completed.

Author Response

Dear colleagues,

We appreciate your comments and suggestions while revising our article. The article has been corrected due to recommendations and we are sending it again.

Make sure of the rows in the article being changed because of correction

Reviewer 4

I believe that references can be completed.

The references of the article have been completed according to the suggestions of the reviser. (References 32-36, Row 106)

Reviewer 5 Report

It is a good work.

I agree that possible instruments an methods were use in teh investigation, but as You now, there is better solution to determin the size distribition of particles in milk products e.g. Malvern mastersizer or other similar instuments. So using this microscopic method was not the best choice. It is not clear that why did you dry the samples for the microstructure analysis... we can explore changes in structure after drying... and not the dried samples were the real end products.

But overall your work will be appropriate for publishing, I think.

Some strong suggestion...

  • I suggest to use not " fat balls", but "fat globules".. or in case of vegetable oils.. fat droplets or particles.
  • You have to perform a serious english revision! Text contains many mistakes!!! Grammatical and spelling errors too. For example lines: 125-126, 134,137,146: (what does it mean.. pure milk fat?), 153 (2 or three times??),  154: what is "natural cream" exactly?  is it coming from milk separation?, 162, 164, 176,205, 267, 281, 408, 450, 454, 463 (vapour) and so on...
  • I do not know exactly but the abbreviation of the dimension of  micrometer (mkm) is not right. But the most important that You have to use same abbreviation in the article.. in the text and also in figures!
  • Some strange word separation can be found in the text.. check them please!
  • The abbreviation of teh coefficient of variation is rahter.. CV  than V. but You hae to use the required form of course... check it!
  • The first and second figures are interchanged!

Author Response

Dear colleagues,

We appreciate your comments and suggestions while revising our article. The article has been corrected due to recommendations and we are sending it again.

Make sure of the rows in the article being changed because of correction

Reviewer 5

I agree that possible instruments an methods were use in teh investigation, but as You now, there is better solution to determin the size distribition of particles in milk products e.g. Malvern mastersizer or other similar instuments. So using this microscopic method was not the best choice. It is not clear that why did you dry the samples for the microstructure analysis... we can explore changes in structure after drying... and not the dried samples were the real end products. But overall your work will be appropriate for publishing, I think. Some strong suggestion... I suggest to use not " fat balls", but "fat globules".. or in case of vegetable oils.. fat droplets or particles. You have to perform a serious english revision! Text contains many mistakes!!! Grammatical and spelling errors too. For example lines: 125-126, 134,137,146: (what does it mean.. pure milk fat?), 153 (2 or three times??), 154: what is "natural cream" exactly? is it coming from milk separation?, 162, 164, 176,205, 267, 281, 408, 450, 454, 463 (vapour) and so on... I do not know exactly but the abbreviation of the dimension of micrometer (mkm) is not right. But the most important that You have to use same abbreviation in the article.. in the text and also in figures! Some strange word separation can be found in the text.. check them please! The abbreviation of teh coefficient of variation is rahter. CV than V. but You have to use the required form of course... check it! The first and second figures are interchanged!

We agree with your remarks and appreciate your comments and suggestions. The text has been properly corrected and revised, the formulas and notations being checked and specified. (Row 75, 91, 139,141, 144, 145, 147, 234, 239, 259, 262, 266, 269, 272, 276 etc.)

Round 2

Reviewer 2 Report

Dear Authors,

I asked you in my note 2 and also in note 7 and in note 10, to make a table with the nomenclature of all the quantities present in the manuscript. In other words, I was asking you for a nomenclature table containing the symbol, description and unit of measurement of each quantity present in the manuscript. A table to be placed at the beginning or at the end of the manuscript that allows the reader to find immediately and centrally this information on the meaning of the symbols present in the manuscript. Instead you have made a "Table 2. The nomenclature of the milk quantities" which contains only the diameter, the number of fat globules and the percentage volume with various values of these quantities. So please prepare a nomenklature table as I  described above.

Author Response

Dear colleagues,

We send you the article with corrections in accordance with the last remark of the reviewer. In particular, in the end of the article a table with nomenclature is added, which contains symbols, their description and units of measurement of each indicator.

Sincerely, the authors.
